# Global Proteomics to Study Silica Nanoparticle-Induced Cytotoxicity and Its Mechanisms in HepG2 Cells

**DOI:** 10.3390/biom11030375

**Published:** 2021-03-02

**Authors:** Sun Young Lee, In Young Kim, Min Beom Heo, Jeong Hee Moon, Jin Gyeong Son, Tae Geol Lee

**Affiliations:** 1Bioimaging Team, Safety Measurement Institute, Korea Research Institute of Standards and Science (KRISS), Daejeon 34113, Korea; sylee12@kriss.re.kr; 2Nano-Safety Team, Safety Measurement Institute, Korea Research Institute of Standards and Science (KRISS), Daejeon 34113, Korea; inyoungkim@kriss.re.kr (I.Y.K.); mbheo@kriss.re.kr (M.B.H.); 3Disease Target Structure Research Center, Korea Research Institute of Bioscience and Biotechnology (KRIBB), Daejeon 34141, Korea; jhdal@kribb.re.kr

**Keywords:** silica nanoparticles, cytotoxicity, label-free quantification, proteomic analysis, mass spectrometry

## Abstract

Silica nanoparticles (SiO_2_ NPs) are commonly used in medical and pharmaceutical fields. Research into the cytotoxicity and overall proteomic changes occurring during initial exposure to SiO_2_ NPs is limited. We investigated the mechanism of toxicity in human liver cells according to exposure time [0, 4, 10, and 16 h (h)] to SiO_2_ NPs through proteomic analysis using mass spectrometry. SiO_2_ NP-induced cytotoxicity through various pathways in HepG2 cells. Interestingly, when cells were exposed to SiO_2_ NPs for 4 h, the morphology of the cells remained intact, while the expression of proteins involved in mRNA splicing, cell cycle, and mitochondrial function was significantly downregulated. These results show that the toxicity of the nanoparticles affects protein expression even if there is no change in cell morphology at the beginning of exposure to SiO_2_ NPs. The levels of reactive oxygen species changed significantly after 10 h of exposure to SiO_2_ NPs, and the expression of proteins associated with oxidative phosphorylation, as well as the immune system, was upregulated. Eventually, these changes in protein expression induced HepG2 cell death. This study provides insights into cytotoxicity evaluation at early stages of exposure to SiO_2_ NPs through in vitro experiments.

## 1. Introduction

Silica nanoparticles (SiO_2_ NPs) are popular materials used in drug delivery and various bio-applications due to their excellent bio-stability, easy surface modification, and ability for fluorescent tagging [1,2,3]. In addition, SiO_2_ NPs are widely used in numerous industries, and can affect people through various routes depending on the manufacturing process [4]. In particular, the enhanced potential for human exposure to engineered SiO_2_ NPs through ingestion, inhalation, or dermal penetration due to their increased prevalence in commercial applications is an important concern [5]. Increasing exposure concerns in the industrial sector have raised global concerns about the safety and potential health impacts of SiO_2_ NPs. According to the Nanomaterials Health Implications Research (NHIR) Consortium, run by the U.S. National Institutes of Environmental Health Sciences, there are three directions for toxicity studies: physicochemical characteristics, in vitro assays (cellular and non-cellular), and in vivo assays [6]. The physicochemical properties of nanoparticles, which were overlooked at the beginning of their toxicity studies, are likely to influence toxicity results. Physicochemical properties, such as particle size and size distribution, agglomeration state, shape, crystal structure, chemical composition, and surface area, could affect the cell and animal responses [7,8]. In general, the smaller size of nanoparticles and more pores on the surface result in higher cytotoxicity [9,10]. SiO_2_ NPs of 50 nm or smaller depending on the treatment time or the presence or absence of serum, but at concentrations above 25 μg/mL, they generally begin to exhibit cytotoxicity [11,12]. When modifying the cell surface, different types of modification ligands can exert different effects in vitro and in vivo. In the case of SiO_2_ NPs produced by the flame synthesis method, which is widely used in the industrial sector, the higher the amount of surface silanol groups, the lower the toxicity shown [13]. In the case of 70 nm SiO_2_ NPs, unmodified SiO_2_ NPs entered near the nucleus more effectively than those modified with amine or carboxyl, and exhibited higher toxicity [14]. Recent studies have shown that cationic nanoparticles induce greater cytotoxicity compared to anionic nanoparticles [15,16,17]. Physicochemical characteristics are an important consideration for assessing toxicity. The confounding effects of poor sample characterization on determining the causes of toxicity necessitates high-quality preparation of nanomaterials for analysis.

SiO_2_ NPs have been subjected to investigations of their biological safety through various methods both in vivo and in vitro. In vivo studies with SiO_2_ NP-treated mice demonstrated the particles were distributed in nearly all organs, and mainly accumulated and induced adverse effects in the lung, spleen, and liver [18,19,20]. The liver is a major organ in detoxification and is well known as a primary target organ for nanoparticles [21]. A recent study of acute exposure to SiO_2_ NPs by Sun et al. demonstrated that liver damage could be induced through macrophage infiltration and granuloma formation, while repeated exposure to SiO_2_ NPs can induce liver fibrosis [22,23]. Many cell lines have been investigated through in vitro studies, and it was reported that excess exposure to SiO_2_ NPs could result in cytotoxic effects [24,25]. It has been shown that SiO_2_ NPs can induce oxidative stress at 3 and 24 h post-treatment in vitro, suggesting that the oxidative stress-mediated mitochondrial pathway may lead to apoptosis, contributing to hepatotoxicity [26]. In addition, it has been reported that Kupffer cells can be activated by SiO_2_ NPs and contribute to hepatotoxicity by releasing bioactive mediators, such as tumor necrosis factor-α, reactive oxygen species (ROS), and nitric oxide [27].

Research evaluating the biological activity and potential risks of nanomaterials has focused on the appropriate selection of biological endpoints [28,29,30]. Unfortunately, the primary focus of many cytotoxicity studies is limited to measuring the direct cytotoxic or pro-inflammatory effects of nanomaterials, rather than attention to subtle changes in biological function [31,32]. Evaluating the interaction of nanomaterials and cells is critical for safety considerations. In particular, mass spectrometry (MS)-based approaches are able to apply deep proteomics analysis for assessing protein adsorption on nanomaterials, proteomic changes, and cytotoxicity occurring when nanomaterials interact with the extracellular matrix through both in vitro and in vivo experiments [33]. Proteomics-based technology offers an attractive approach to both unbiased and multivariate systems analysis in evaluating nanomaterial and biological model interactions [34]. Recently, proteomic approaches have been used to evaluate the potential cytotoxicity of 11 different metal and metal oxide nanoparticles through quantitative cellular proteome profile studies [35]. In addition, the qualitative characterization of corona proteins on nanomaterials has become possible [36]. In the present study, we aimed to investigate the biological mechanisms underlying cell death following initial nanoparticle exposure using proteomics. As the proteome of the cell changes during the initial stage of exposure to nanoparticles, prior to the occurrence of any morphological changes, proteomic analysis based on MS was expected to provide insights into early changes underlying nanoparticle-induced cytotoxicity.

## 2. Materials and Methods

### 2.1. Materials and Reagents

For 20 nm SiO_2_ NPs (301-01-002), the certified reference materials made by Korea Research Institute of Standards and Science were used. Tris-HCl buffer (pH 8.0), phosphate buffered saline (PBS), sodium chloride (NaCl), formic acid (FA), ammonium bicarbonate (AmBic), dithiothreitol (DTT), iodoacetamide (IAA), and L-cysteine were purchased from Sigma-Aldrich (St. Louis, MO, USA). Protease inhibitor cocktail were purchased from Roche Diagnostic GmbH (Mannheim, Germany). In order to digest the proteins, trypsin was purchased from Promega (Madison, WI, USA). HLB cartridge purchased from Waters (Milford, MA, USA) was used. Water (with 0.1% FA), acetonitrile (ACN) (with 0.1% FA), n-dodecyl beta-D-maltoside (DDM), and bicinchoninic acid (BCA) protein assay reagent were purchased from Thermo Fisher Scientific (Rockford, IL, USA).

### 2.2. Nanoparticles Characterization

The shape and sizes on the surfaces of SiO_2_ NPs were conducted by using transmission electron microscopy (TEM, TECNAI G2 F30 S-Twin), which was operated at 300 keV. The FT-IR spectrum was obtained using a Nicolet iS10 FT-IR spectrometer (Thermo Fisher Scientific) equipped with an attenuated total reflectance (ATR) accessory (Smart Miracle, PIKE Tech). One μL of SiO_2_ NPs solution was placed on a ZnSe-ATR crystal and dried under vacuum for 2 h. The HgCdTe detector cooled by liquid N_2_ was used to collect the reflected light. A total of 16 scans were averaged to yield a spectrum at 4 cm^−1^ resolution. The average hydrodynamic size and distribution of the NPs in water were determined by using dynamic light scattering (DLS, Nano ZS90, Malvern Instruments Ltd., Worcestershire, UK). All measurements were conducted in disposable cuvettes and the samples were analyzed with a 4-mW laser operating at a wavelength of 633 nm at 25 °C and the scattering angle was fixed at 90°. The surface charge (zeta potential) of the NPs in water was detected by using an electrophoretic light scattering (ELS, Nano ZS90, Malvern Instruments Ltd., Worcestershire, UK). All samples were prepared by diluting the stock in deionized (DI) water. The operating temperature was kept constant at 25 °C.

### 2.3. HepG2 Cell Culture and SiO_2_ NPs Treatment

The HepG2 cells were purchased from the American Type Culture Collection (ATCC, Manassas, VA, USA) and maintained in Dulbecco’s modified eagle’s medium (DMEM) added with 10% fetal bovine serum (FBS) and 1% penicillin-streptomycin (GIBCO-BRL, Grand Island, NY, USA). The cells were incubated in 5% CO_2_ at 37 °C. For the treatment of cells, SiO_2_ NPs were dispersed as 30 μg/mL concentration in a serum-free DMEM. The HepG2 cells were seeded and incubated for 24 h. After the cells were washed with PBS, it was treated with prepared SiO_2_ NPs for 4, 10, or 16 h.

### 2.4. Cell Viability Assay

To evaluate the cell viability, we measured lactate dehydrogenase (LDH) release using CytoTox96^®^ Non-Radioactive Cytotoxicity Assay (Promega, Madison, WI, USA). After the cells were treated as indicated above, 100 μL of supernatant was collected and tested according to the manufacturer’s instructions.

### 2.5. ROS Detection and Quantitation

SiO_2_ NPs treated cells were washed in PBS and exposed 10 μM CM-H_2_DCF-DA (DCF) for 30 min at 37 °C. After washing again with PBS, the fluorescence of DCF was measured using a microplate reader.

### 2.6. Sample Preparation for Proteomics

In order to obtain the protein lysates from the HepG2 cells, 150 μL of lysis buffer composing of 0.2% DDM, 150 mM NaCl, and one tablet of protease inhibitor cocktail in 50 mM Tris-HCl solution was added to the cell pellets. The cell mixture was vortexed for 1 min, and then incubated at 4 °C for 1 min. The solution was centrifuged at 12,000× *g* for 15 min at 4 °C, the supernatants were transferred into new tubes for the isolation of proteins. The concentration of protein lysates was measured by using a BCA protein assay.

Twenty microgram of proteins was reduced by adding 50 mM AmBic buffer containing 10 mM DTT for 2 h at 37 °C, and were subsequently alkylated in the dark for 30 min at room temperature, after the addition of 20 mM IAA solution. In order to remove the remaining IAA, 40 mM L-cysteine was added and was allowed to incubate for 30 min at room temperature, followed by tryptic digestion of the samples (enzyme to substrate ratio of 1:20). The digestion was carried out overnight at 37 °C, and quenched with 5% FA. The digests were desalted by HLB cartridge and eluted with 0.5 mL H_2_O:ACN (50:50, *v*/*v*) solution containing 0.1% FA. After evaporation, the peptides were stored at −80 °C prior to nanoflow liquid chromatography-electrospray ionization-tandem mass spectrometry (nanoLC-ESI-MS/MS) analysis.

### 2.7. Nanoflow Liquid Chromatography Tandem Mass Spectrometry

The dried samples were reconstituted with H_2_O:ACN (98:2, *v*/*v*) solution containing 0.1% FA, and 500 ng of peptide mixture was used for proteomic analysis. The peptide mixture was analyzed by using a NanoElute LC system connected to a hybrid trapped ion mobility spectrometry-quadrupole time-of-flight mass spectrometer (timsTOF Pro, Bruker Daltonics, Bremen, Germany), equipped with a modified nano-electrospray ion source (CaptiveSpray, Bruker Daltonics, Billerica, MA, USA). The peptide mixtures were separated at 50 °C with a constant flow of 400 nL/min on a homemade column (75 μm—inner diameter, 250 mm—length) packed with C18 resins (1.9 μm, 120 Å, Dr. Maisch, Ammerbuch, Germany), and eluted with the following binary gradient of mobile phases A (0.1% FA in H_2_O) and B (0.1% FA in ACN): 2% to 17% B for 45.0 min, 17% to 25% for 22.5 min, 25% to 37% for 7.5 min, 37% to 80% for 5.0 min, and then maintained for 10 min to rinse the analytical column. The timsTOF Pro was operated in PASEF mode using Compass Hystar 5.0.37.1. Settings for MS and MS/MS scans were as follows: mass range 100 to 1700 *m*/*z*, 1/K_0_ start 0.6 V⋅s/cm^2^ end 1.6 V⋅s/cm^2^, capillary voltage 1400 V, dry gas 3 L/min, dry temp 180 °C; PASEF mode: 10 MS/MS scans (total cycle time 1.16 s), charge range 0–5, active exclusion for 0.4 min, scheduling target intensity 20,000, intensity threshold 2500, and CID collision energy 20–59 eV, depending on precursor mass and charge.

### 2.8. Data Analysis

The obtained raw data were submitted to PEAKS Studio 10.5 (Bioinformatics Solutions, Waterloo, ON, Canada) to search against the SwissProt database of *Homo sapiens* (human, UP00000564, downloaded 22/11/2019, 20379 entries) from Uniprot (www.uniprot.org/ (accessed on 20 January 2021)) with a false discovery rate (FDR) of 0.01 for protein identification and label free quantification (LFQ). The search parameters for identification were: (a) trypsin as specific enzyme, two missed cleavage allowed; (b) fixed modification: carbamidomethylation of cysteine and variable modification: oxidation of methionine and acetylation of protein N-term, allowing for three variable PTM per peptide; (c) precursor mass error tolerance of 20.0 ppm; (d) fragment mass error tolerance of 0.05 Da. When protein identification was completed, LFQ was carried out using the analyzed PEAKS file. LFQ analysis was performed by analysis of variance (ANOVA) method, and significance thresholds were set to two unique peptides, a data filter in at least three samples per group, a significance of 20 (*p*-value = 0.01), and a 2.0-fold change. Normalization of data was conducted using total ion chromatography (TIC). The resulting data were exported to Microsoft Excel, and a Venn diagram was generated by using an online tool, Venny v2.1 (https://bioinfogp.cnb.csic.es/tools/venny/ (accessed on 20 January 2021)). Heatmap and principal component analysis (PCA) were performed using the Perseus v1.5.8.5.

## 3. Results

### 3.1. Silica Nanoparticle Characterization

For quantitative proteomics of SiO_2_ NP-induced cytotoxicity, 20 nm SiO_2_ NPs were used. The size and morphology characterizations were conducted by using DLS and TEM, respectively. TEM showed that the SiO_2_ NPs have a spherical morphology with a small size distribution (Figure 1A). In the pristine state, the 20-nm SiO_2_ NPs had a diameter of 19.6 nm (±0.5) when measured by TEM. Hydrodynamic size and zeta potential of the used SiO_2_ NPs were 21.0 nm (±0.1) and −34.3 mV (±3.5), respectively (Table 1). The FT-IR spectrum of 20 nm SiO_2_ NPs is shown in Figure 1B. In the spectral region, the band at 1080 cm^-1^ and 804 cm^-1^ are due to the asymmetric and symmetric stretching vibration of Si–O–Si. The band at 973 cm^-1^ confirms the existence of Si–OH stretching vibration [37]. The wide absorption band around 3600–3200 cm^-1^ is attributable to the stretching of the OH groups from the physically absorbed water [38]. It has been shown that the 20 nm SiO_2_ NP surfaces used in this study are very clean without any organic material.

### 3.2. Cell Viability and LDH Leakage Assays

To evaluate the toxic effect of SiO_2_ NPs on HepG2 cells, bright-field microscopy and the LDH leakage assay were conducted. In the upper images of Figure 2A, no significant change was observed in the control groups from 0 to 16 h. Following 4 h of exposure to SiO_2_ NPs (4 h SiO_2_ NP group), the cellular morphology of HepG2 cells did not change (Figure 2A), while the LDH level increased for SiO_2_ NP-treated groups compared with the levels in the 0 h control group (Figure 2B). The morphological changes were observed in the microscopic data with increasing time after 10 h exposed to SiO_2_ NPs (10 h SiO_2_ NP group); these results corresponded to the increased membrane damage measured by LDH leakage assay. LDH levels 16 h (16 h SiO_2_ NP group) after exposure to SiO_2_ NPs increased by 50% compared to the 0 h control group.

### 3.3. Proteomic Analysis

To investigate the cellular response to SiO_2_ NP exposure through changes in protein levels, we performed LFQ of SiO_2_ NP-treated HepG2 cells according to each elapsed time (4, 10, and 16 h) in comparison with that of each control group. Comparing the results of the 4, 10, and 16 h SiO_2_ NP groups with each control group, a total of 2129, 2504, and 2544 proteins were identified, respectively (Figure 3A). In the three groups, 1937 common proteins were observed. Non-overlapping proteins were also observed in groups of 61, 142, and 189 in 4, 10, and 16 h groups, respectively. Sixty-one proteins exclusive to the 4 h group are associated with mitochondria, while 142 and 189 proteins of the 10 and 16 h groups, respectively, are related to cytosol, cytoplasm, and organelles (Appendix A). These results suggest that the biological responses of SiO_2_ NPs in HepG2 cells begin in the mitochondria and subsequently occur in the cytoplasm, then the cytosol, followed by the organelles.

We confirmed the trend in cellular response over time through the Venn diagram analysis (Figure 3A). To assess the changes according to time in more detail, proteins which were upregulated and downregulated were extracted from each SiO_2_ NP group compared to each control group (Figure 3B). According to the SiO_2_ NP-treated group and the control group for the 4 h comparison result, 275 downregulated and 53 upregulated proteins were observed. In HepG2 cells, 149 downregulated and 42 upregulated proteins were detected as the SiO_2_ NPs treatment time elapsed for 10 h, but the number of the differentially expressed proteins over exposure of SiO_2_ NPs after 16 h were increased (236 downregulated and 76 upregulated proteins). We attempted to analyze the time course of cytotoxicity by constructing a protein–protein interaction network using the STRING algorithm for differently expressed proteins between time groups. Following 4 h of exposure to SiO_2_ NPs, downregulated proteins were related to the ribosome, mRNA splicing, RNA transport, and the cell cycle (Figure 4A). Some of these proteins which were involved in ribosomes, mRNA splicing, and cell cycle were shown to be continuously downregulated even in the 10 h (Figure 4C) and 16 h SiO_2_ NP groups (Figure 4E). The 53 proteins upregulated in the 4 h group were associated with immune system and DNA damage/telomeres stress-induced aging (Figure 4B), and the same interaction was found at 10 h (Figure 4D). In the 16 h SiO_2_ NP group, upregulated proteins involved in the immune system and DNA damage/telomere stress-induced senescence were found to be increased, and proteins related to oxidative phosphorylation were detected (Figure 4F). In previous studies, it has been reported that oxidative phosphorylation proteins were upregulated under the influence of ROS generated from SiO_2_ NPs [39,40,41]. To determine whether these proteins were indeed affected by ROS production in our system, H_2_-DCF-DA staining was performed to measure ROS levels in HepG2 cells in accordance with the elapsed time after exposure to SiO_2_ NPs. There was no significant change even after exposure to SiO_2_ NPs for 4 h compared to the 0 h control group; however, after 10 h, it was confirmed that the ROS level significantly increased with elapsed time (Appendix A). Based on these results, we demonstrated that ROS produced by SiO_2_ NP-induced cellular responses in HepG2 cells such as oxidative phosphorylation and immune system. Detailed information for down- and upregulated proteins involved in specific interactions as indicated in Figure 4 is described in Appendix A, respectively.

To investigate changes to proteins involved in specific biological pathways over the time of exposure to SiO_2_ NPs in HepG2 cells, four control groups (0, 4, 10, and 16 h) and three experimental groups (4, 10, and 16 h SiO_2_ NP groups) simultaneously underwent LFQ-based proteomic analysis. From the proteomic analysis, the result was confirmed the intracellular effects generated from both serum-free and the SiO_2_ NPs. Considering that the proteins level did not change significantly in the control group, whereas the proteins were upregulated and downregulated in the SiO_2_ NPs group, it can be seen that the significant changes on HepG2 cells resulted from SiO_2_ NPs rather than a serum-free environment. A total of 1668 proteins were quantified through the analysis, and 482 proteins were differentially expressed with correlations among the seven groups. As a result of drawing heat maps of 482 proteins with correlations among the seven groups, 363 downregulated (green color) and 119 upregulated proteins (red color) were quantified compared to 0 h control (Figure 5A). Principal component analysis was performed using the differentially expressed proteins, which revealed that the principal components distinctly differed among the SiO_2_ NP-treated groups compared to control groups (Figure 5B). With regards to the first component, the largest difference was observed between the controls and SiO_2_ NP groups. In addition, it was confirmed that the SiO_2_ NPs treated group can be classified by time through principal component 2. Based on these results, it was possible to understand the cellular response over time after exposure to SiO_2_ NPs by assessing the changes in protein levels.

To explore the detailed cellular responses to SiO_2_ NP exposure, we investigated the specific pathways associated with the differentially expressed proteins among the six groups compared to the 0-h control. As shown in Table 2, mitochondrial-related proteins, such as cell cycle and apoptosis regulator protein 2 (CCAR2), vacuolar protein sorting-associated protein 35 (VPS35) and dynamin-1-like protein (DNM1) were downregulated after exposure to SiO_2_ NPs, whereas these proteins in control groups were unchanged and VPS35 in the 16 h control was upregulated. These results indicate that SiO_2_ NPs introduced into cells have a direct effect on mitochondria. The exposure of SiO_2_ NPs in HepG2 cells affects the regulation of proteins involved in the cell cycle including nucleoprotein TPR (TPR), ubiquitin-conjugating enzyme E2 N (UBE2N), and cyclin-dependent kinase 1 (CDK1) [42]. In addition, proteins associated with DNA replication, such as DNA polymerase delta subunit 3 (DPOD3), ATP-dependent RNA helicase A (DHX9), and RNA-binding protein 14 (RBM14) were more downregulated in the 4 h group of HepG2 cells with SiO_2_ NPs. Likewise, the most downregulated pathway in the 4 h SiO_2_ NP group was related to spliceosomes, and the proteins involved were measured for SNW domain-containing protein 1 (SNW1), RNA-binding protein 25 (RBM25), and Probable ATP-dependent RNA helicase DDX46 (DDX46). This result may suggest that protein–protein interactions associated with mRNA splicing in Figure 4 occurs via spliceosomes. Finally, carbon metabolism-related proteins, such as alpha-enolase (ENOA), cytoplasmic aconitate hydratase (ACOC), and ATP-dependent 6-phosphofructokinase muscle type (PFKAM), were also downregulated by SiO_2_ NPs.

We also investigated the major cellular pathways associated with the proteins observed to have been upregulated, as listed in Table 3. Among the upregulated proteins, endosome-related proteins, including anthrax toxin receptor 1 (ANTR1), thioredoxin domain-containing protein 5 (TXND5), and CD81 antigen (CD81), were significantly upregulated 4 h after exposure to SiO_2_ NPs, and continued to be upregulated up to 16 h. Proteins associated with the phagosome, such as HLA class I histocompatibility antigen A alpha chain (HLAA), integrin beta-1 (ITB1), and V-type proton STPase subunit d1 (VA0D1), were upregulated in the 10 h SiO_2_ NP group; most of these were upregulated to a greater extent with increasing exposure time to SiO_2_ NPs. When extrinsic substances are introduced into cells by endosomes and phagosomes, the production of lysosomes is activated to decompose these substances. Thus, it can be seen that some of the proteins associated with the lysosome, such as dolichyl-diphosphooligosaccharide-protein glycosyltransferase 48 kDa subunit (OST48), lysosome-associated membrane glycoprotein 2 (LAMP2), and prenylcysteine oxidase 1 (PCYOX), were upregulated in the 4 h SiO_2_ NP group, and most of the proteins were significantly upregulated in the 10 h SiO_2_ NP group. Since endosomes and phagosomes continue to form until an elapsed time of 16 h, it was observed that proteins related to lysosomes were significantly upregulated even in the 16 h SiO_2_ NP group. Unlike carbon metabolism, most proteins involved in lipid metabolism, including lysophospholipid acyltransferase 7 (MBOA7), sphingolipid delta(4)-desaturase DES1 (DEGS1), and medium-chain specific acyl-CoA dehydrogenase (ACADM), were significantly upregulated in the 10 and 16 h SiO_2_ NP groups. Some proteins are upregulated in the control without SiO_2_ NP, but since these proteins are further upregulated in the presence of SiO_2_ NP, it was confirmed that several intracellular pathways are affected by SiO_2_ NP. In Figure 6, MS and tandem MS spectra for four representative proteins involved in the biological pathways are shown. The representative downregulated [CDK1 (LESEEEGVSTAIR) and ACOC (QAPQTIHLPSGEILDVFDAAER)] and upregulated proteins [ACADM (IYQIYEGTSQIQR) and LAMP2 (GILTVDELLAIR)] were also found.

## 4. Discussion

As SiO_2_ NPs are widely used in the field of nanomedicine, food additives, and consumer products, detailed studies into their safety and biological toxicity are required. SiO_2_ NPs are easily introduced into cells at a size of 50 nm or lower and are known to cause toxicity when used at a concentration of 25 μg/mL or higher after 24 h in vitro [11,12,43]. Recent studies have demonstrated various pathways for SiO_2_ NP-induced apoptosis, such as DNA damage [44,45,46], lysosome-induced cell death [47], and mitochondrial dysfunction [48,49]. However, most of these studies are the results of cellular responses following at least 24 h of exposure to these nanoparticles. It is known that uptake of the nanoparticles begins as rapidly as 15 min after cellular exposure, and this uptake is associated with several intracellular reactions leading to cytotoxicity [43]. In this study, we investigated the biological reaction to these nanoparticles at multiple time points (0, 4, 10, and 16 h) after cellular entry of SiO_2_ NPs compared to cells not treated with SiO_2_ NPs. Furthermore, we investigated the molecular mechanisms underlying their cytotoxicity through LFQ-based proteomic analysis. Interestingly, analysis of differentially expressed proteins in response to SiO_2_ NP exposure led to a clear understanding of a temporal pattern in cellular response to 20 nm SiO_2_ NPs and their cytotoxic potential. The protein profiling approach is a useful tool to determine how cells induce cytotoxicity when exposed to nanoparticles and how these changes ultimately result in cell death, as overall changes in protein levels be detected even in the absence of significant extracellular changes.

Exogenous substances such as nanoparticles enter cells through endocytic and phagocytic pathway [50,51]. In our study, the SiO_2_ NPs were firstly introduced into HepG2 cells through endocytic pathway, considering that endocytosis-related proteins were upregulated in the 4 h SiO_2_ NP groups. In particular, the increase in Flotullin-2 (FLOT2) is expected to cause flotillin-dependent endocytosis [52]. Phagosome-related proteins were upregulated at 10 h, which is speculated that introduced into cells after an aggregation due to exposure of the nanoparticles to the media. The endosome and phagosome formed in the cell are combine with the lysosomes to form endolysosomes and phagolysosomes, respectively. These complexes containing the ingested SiO_2_ NPs fuse with lysosomes are then decomposed by enzyme-catalyzed hydrolysis [50]. Some of the lysosome-related proteins were upregulated in the 4 h group, and many proteins related to lysosomes were continuously upregulated from 10 to 16 h following exposure to SiO_2_ NPs. These results demonstrated that when HepG2 cells are exposed to SiO_2_ NPs, endocytosis occurs within 4 h, phagocytosis occurs at 10 h, and lysosomal proteins involved in the decomposition of foreign substances begin to activate from 4 h and continue to be upregulated until 16 h. HepG2 cells exposed to silica nanoparticles were observed to downregulate proteins related to ribosomes and mRNA splicing in the 4 h group. Changes in these proteins are known to induce DNA damage when exogenous damaging substances are introduced into cells [53,54]. In many studies, DNA damage has been reported to be caused by ROS after exposure to nanoparticles [11,25,55]. However, in this study, ROS was found to increase 10 h after exposure to SiO_2_ NPs. These results suggest that the downregulation of proteins associated with mRNA splicing and ribosomes within 4 h in HepG2 cells could be attributed to DNA damage caused by SiO_2_ NPs directly, not ROS. In addition, downregulated proteins related to mRNA splicing and ribosomes subsequently affected DNA replication and cell cycle arrest [53]. In particular, important factors in the regulation of the G2/M phase of the cell cycle is CDK1. Downregulation of G2/M phase-related proteins induces cell cycle arrest, eventually leading to cell death. These results are consistent with previous studies involving cell cycle arrest when cells were exposed to SiO_2_ NPs [11,42,56,57].

Previous studies have reported mitochondrial damage caused by nanoparticles or ROS as a major cause of SiO_2_ NPs cytotoxicity [40,41,44,48]. In our results, we confirmed that mitochondria-related proteins were significantly downregulated in the 4 h SiO_2_ NP group. These results suggest that the factor inducing mitochondrial damage in HepG2 cells is SiO_2_ NPs itself. The nanoparticles introduced into the cells activated the immune system from the beginning of the introduction and caused DNA damage, in which the tendency remained the same even in the 10 h groups. After 16 h of exposure to SiO_2_ NPs, oxygen molecules produced by ROS activate oxidative phosphorylation by upregulating cytochrome b-c1 complex subunit Rieske (UQCRFS1), cytochrome c oxidase subunit 2 (MT-CO2), NADH-ubiquinone oxidoreductase chain 4 (MT-ND4), and succinate dehydrogenase cytochrome b560 subunit (SDHC). Furthermore, the proteins involved in lipid metabolism such as DEGS1, MBOA7, and ACADM were also upregulated in the 10 and 16 h groups. Duan et al. confirmed that the proteins involved in hepatic lipid metabolism were upregulated when exposed to SiO_2_ NPs; these results are consistent with our results [21]. In contrast, the carbon metabolism-related proteins, which regulate energy production, were downregulated in the 4 h group. Lee et al. demonstrated that exposure to SiO_2_ NPs in human embryonic kidney 293 (HEK293) cells resulted in disorders of glucose metabolism as well as glucose uptake [58,59]. Gradually, as the exposure time of SiO_2_ NPs increases, cellular responses such as lipid metabolism and oxidative phosphorylation increase, and lead to the activation of the immune system. The analysis of intracellular changes in HepG2 that occurred 16 h after exposure to SiO_2_ NPs allowed a closer look at the mechanisms of early cytotoxicity. When SiO_2_ NPs are exposed to cells for longer periods, the cells eventually die through a cytotoxic pathway, such as autophagy [60,61], apoptosis [9,62,63], lysosome-induced cell death [47,50], and necrosis [10,30,64] (Figure 7). Although this study focused on global changes in the early stages of nanoparticle exposure, we expect that proteomic analysis following a future separate organelle-based study will enable a deeper understanding of the molecular processes underlying their cytotoxicity and eventuating in SiO_2_ NP-induced cell death. Additionally, the complementary research with a biological assay will provide a more detailed mechanism and biological pathway at a specific time point.

## 5. Conclusions

In this study, cytotoxicity was investigated in HepG2 cells according to the elapsed time (0, 4, 10, and 16 h) of exposure to SiO_2_ NPs. Although the morphology of cells did not change at the early stages (0 to 4 h), membrane damage at 4 h by LDH leakage assay was confirmed. To investigate a detailed mechanism of HepG2 cell responses through the uptake of SiO_2_ NPs, cells were harvested according to the above time intervals compared with cells not treated with SiO_2_ NPs, and proteomic analysis was performed using mass spectrometry. LFQ-based proteomic analysis results showed that SiO_2_ NPs could be internalized into cells through several pathways, such as endocytosis and phagocytosis, resulting in an increase in the number of lysosomes. In addition, cell metabolism and cell division, such as mRNA splicing, cell cycle arrest, and mitochondrial dysfunction, already begin at 4 h after exposure to SiO_2_ NPs. ROS measurement using H_2_-DCF-DA staining showed a significant increase in ROS levels at 10 h, which is consistent with the morphological changes. This finding suggests that DNA damage caused changes directly in HepG2 cells by SiO_2_ NPs rather than by ROS. By 10 to 16 h, ROS activated intracellular oxidative phosphorylation and the immune system of HepG2 cells. These results explain the mechanism of hepatotoxicity following exposure to SiO_2_ NPs. The mass spectrometry-based proteomic approach has been suggested as a global evaluation method to confirm the cytotoxicity and biological effect of SiO_2_ NPs. This approach will also help to understand the cytotoxicity mechanisms of various nanomaterials.

## Figures and Tables

**Figure 1 biomolecules-11-00375-f001:**
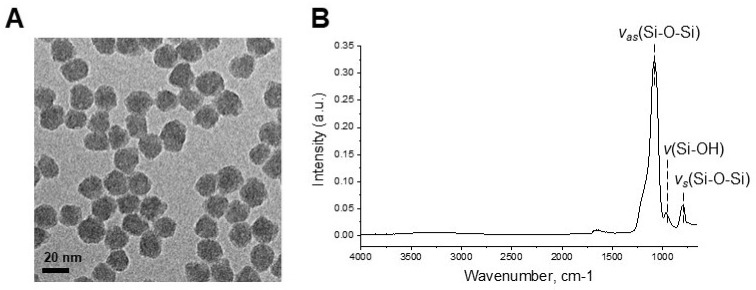
(**A**) Transmission electron microscopic image and (**B**) FT-IR spectrum of 20 nm SiO_2_ NPs before culturing with HepG2 cells.

**Figure 2 biomolecules-11-00375-f002:**
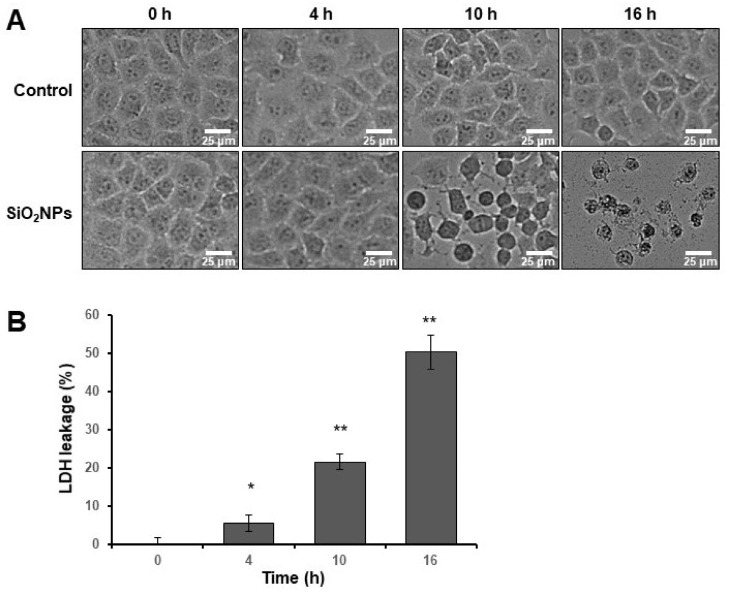
Cell viability assay. (**A**) The morphologies of HepG2 cells using bright-field microscopy. The upper images are HepG2 cells without SiO_2_ NPs, and the lower images are SiO_2_ NP (30 μg/mL)-treated groups exposed for 0, 4, 10, and 16 h. (**B**) Lactate dehydrogenase (LDH) leakage assay of SiO_2_ NP-treated groups with increasing exposure time. Data shown are mean ± standard deviation of three independent experiments. Asterisks indicate effects in comparison to the 0 h control group (* *p* < 0.05, ** *p* < 0.005).

**Figure 3 biomolecules-11-00375-f003:**
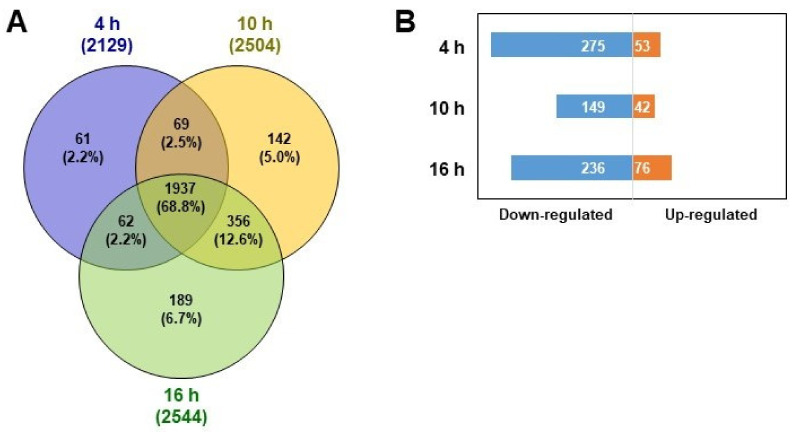
Label-free quantification (LFQ)-based proteomic analysis of HepG2 cells in comparison with controls and SiO_2_ NP groups at each time point (4, 10, and 16 h). (**A**) Venn diagram of the quantified proteins after the treatment of SiO_2_ NPs for 4, 10, and 16 h. (**B**) Upregulated and downregulated proteins in HepG2 cells exposed to SiO_2_ NPs after 4, 10, and 16 h compared to their control groups.

**Figure 4 biomolecules-11-00375-f004:**
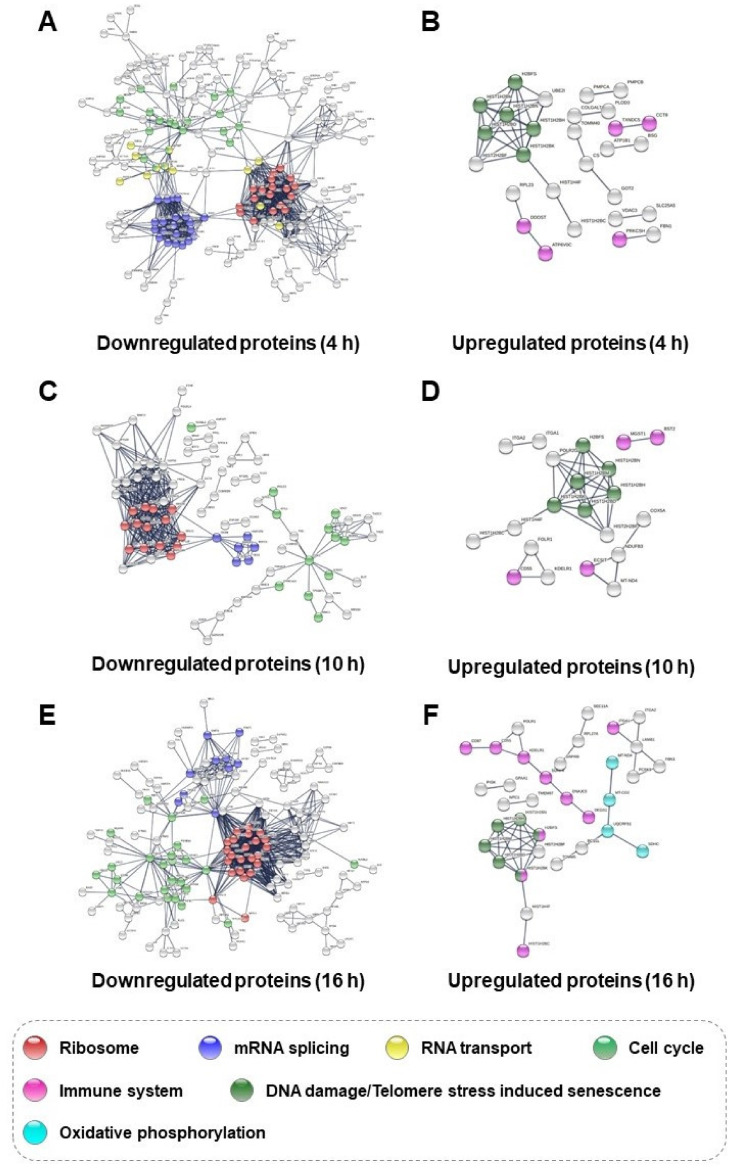
Protein–protein interaction networks (PPI) of differentially expressed proteins in HepG2 cells regulated by SiO_2_ NPs using a STRING algorithm. The color circles represent the cellular interaction with highest confidence (score ≥ 0.9) are shown. PPI of downregulated and upregulated proteins for (**A**,**B**) 4, (**C**,**D**) 10, and (**E**,**F**) 16 h groups.

**Figure 5 biomolecules-11-00375-f005:**
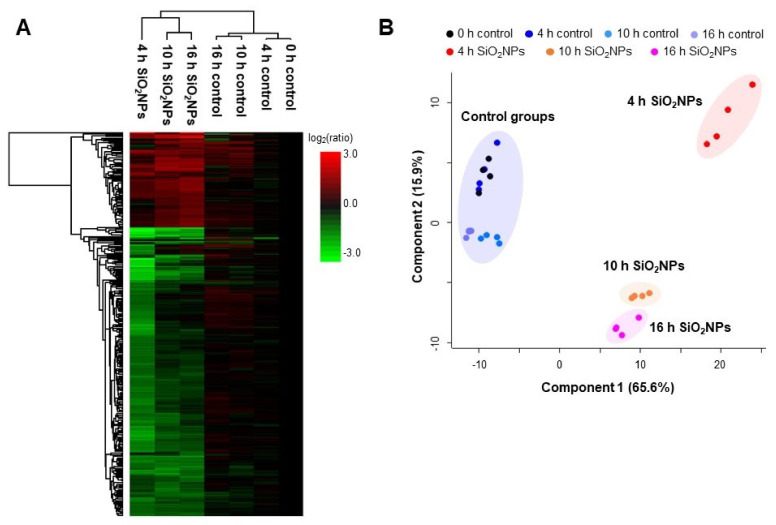
(**A**) Heatmap (with hierarchical cluster analysis) and (**B**) principal component analysis of the differentially expressed proteins from SiO_2_ NPs-treated HepG2 cells in increase of exposure time in comparison to the 0 h group. The filled circles of the same color represent four duplicates.

**Figure 6 biomolecules-11-00375-f006:**
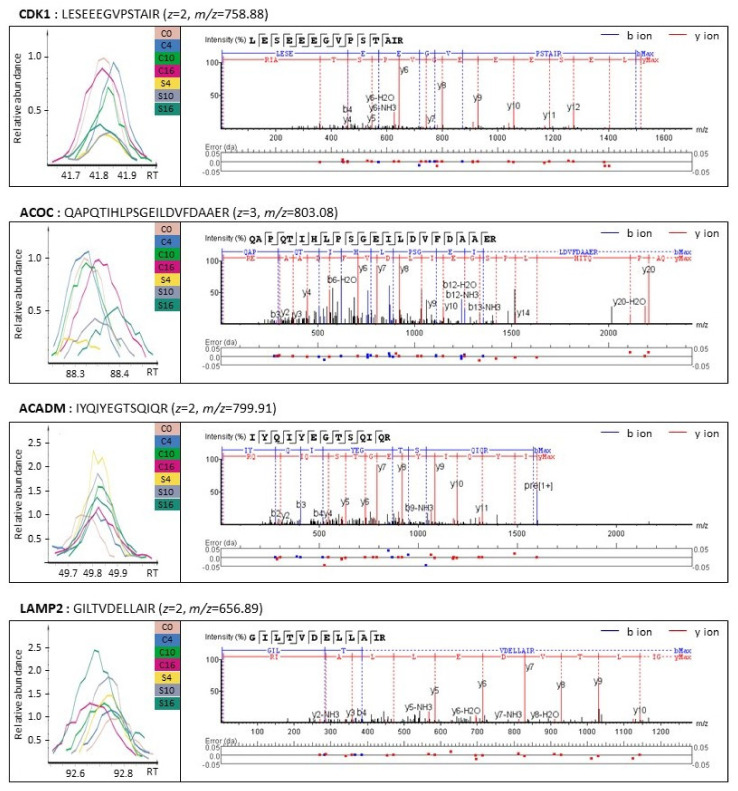
Representative MS and tandem MS spectra of downregulated (CDK1 and ACOC) and upregulated proteins (ACADM and LAMP2) in SiO_2_ NP-treated groups, which are involved in cell cycle checkpoint, carbon metabolism, lipid metabolism, and lysosome. Data were obtained using the software PEAKS Studio 10.5. (Abbreviations: CDK1, cyclin-dependent kinase 1; ACOC, cytoplasmic aconitate hydratase; ACADM, medium-chain specific acyl-CoA dehydrogenase mitochondrial; LAMP2, lysosome-associated membrane glycoprotein 2).

**Figure 7 biomolecules-11-00375-f007:**
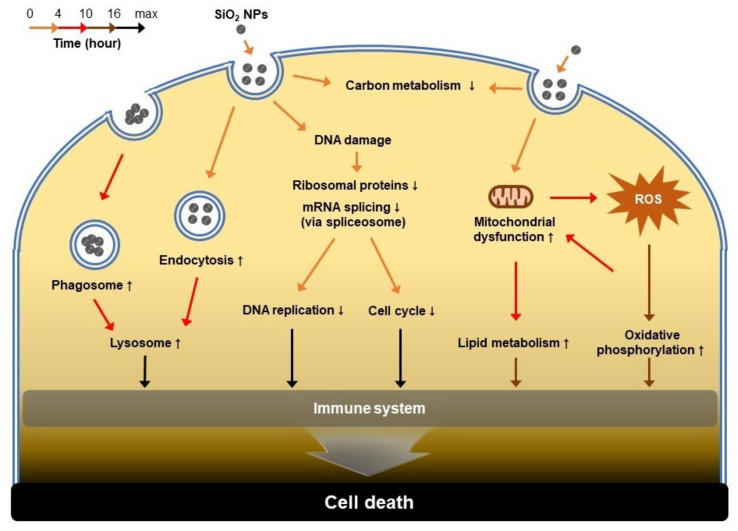
Mechanisms of SiO_2_ NP-induced cytotoxicity in HepG2 cells. The order of biological reactions that occur when SiO_2_ NPs are exposed to cells is indicated by arrows. Orange, 4 h; red, 10 h; brown, 16 h; black, after 16 h. Responses observed in this study are indicated in bold letters, and those described through reference studies are indicated in light letters.

**Table 1 biomolecules-11-00375-t001:** Characterization of SiO_2_ NPs (Mean ± SD, *n* = 3).

Characterization Techniques	20 nm SiO_2_ NPs
TEM (nm)	19.6 ± 0.5
DLS (nm)	21.0 ± 0.1
Polydispersity Index	0.139 ± 0.011
Zeta potential (mV)	−34.2 ± 3.5

**Table 2 biomolecules-11-00375-t002:** List of downregulated proteins from control and SiO_2_ NP groups of HepG2 cells by the exposure time associated with the biological pathway.

Protein	Description	Fold Change Compared to 0 h Control						*p*-Value
		Control (h)			SiO_2_ NPs (h)			
		4	10	16	4	10	16	
**Mitochondria**								
CARE	Cell cycle and apoptosis regulator protein 2	1.06	0.96	1.25	0.54	0.38	0.55	2.09 × 10^−7^
VPS35	Vacuolar protein sorting-associated protein 35	1.07	1.32	1.51	0.39	0.88	0.91	3.52 × 10^−7^
DNM1L	Dynamin-1-like protein	1.04	1.08	1.03	0.47	0.77	0.66	2.19 × 10^−5^
**Cell Cycle Checkpoint**								
TPR	Nucleoprotein TPR	1.06	1.14	1.12	0.39	0.52	0.57	1.32 × 10^−9^
UBE2N	Ubiquitin-conjugating enzyme E2 N	0.94	1.20	1.19	0.36	0.68	0.62	6.41 × 10^−9^
PRKDC	DNA-dependent protein kinase catalytic subunit	1.04	1.11	1.17	0.42	0.63	0.84	1.25 × 10^−8^
MDC1	Mediator of DNA damage checkpoint protein 1	0.93	1.00	1.11	0.39	0.50	0.54	1.48 × 10^−8^
PCH2	Pachytene checkpoint protein 2 homolog	1.11	1.35	1.20	0.18	0.56	0.78	6.76 × 10^−8^
PIP30	PSME3-interacting protein	1.00	0.88	0.82	0.47	0.44	0.44	6.84 × 10^−8^
CDK1	Cyclin-dependent kinase 1	1.01	0.94	1.13	0.38	0.45	0.45	8.97 × 10^−8^
NSUN2	RNA cytosine C(5)-methyltransferase NSUN2	1.02	0.85	0.94	0.45	0.38	0.41	1.14 × 10^−7^
RCC2	Protein RCC2	1.05	1.09	1.14	0.44	0.71	0.78	1.23 × 10^−7^
MARE1	Microtubule-associated protein RP/EB family member 1	1.00	1.12	1.26	0.31	0.75	0.68	6.67 × 10^−7^
NU107	Nuclear pore complex protein Nup107	0.97	0.97	1.18	0.44	0.67	0.66	1.08 × 10^−6^
MRE11	Double-strand break repair protein MRE11	1.02	1.20	1.47	0.41	0.52	0.72	1.11 × 10^−6^
PSD13	26S proteasome non-ATPase regulatory subunit 13	1.04	0.82	0.86	0.47	0.58	0.50	1.82 × 10^−6^
PTN11	Tyrosine-protein phosphatase non-receptor type 11	1.10	1.24	1.16	0.48	0.76	0.60	4.69 × 10^−6^
PCNA	Proliferating cell nuclear antigen	1.07	1.01	1.35	0.37	0.61	0.74	1.08 × 10^−5^
TP53B	TP53-binding protein 1	0.67	1.07	0.94	0.18	0.32	0.44	4.89 × 10^−5^
ZW10	Centromere/kinetochore protein zw10 homolog	0.90	1.13	0.84	0.25	0.78	0.96	8.63 × 10^−3^
**DNA Replication**								
THOC4	THO complex subunit 4	0.99	0.99	1.03	0.39	0.55	0.56	1.08 × 10^−12^
NUCKS	Nuclear ubiquitous casein and cyclin-dependent kinase substrate 1	0.98	1.03	1.12	0.23	0.36	0.48	2.71 × 10^−11^
MCM6	DNA replication licensing factor MCM6	1.12	0.96	1.31	0.32	0.47	0.58	7.11 × 10^−11^
DHX9	ATP-dependent RNA helicase A	1.05	1.16	1.13	0.44	0.57	0.65	4.08 × 10^−8^
MCM3	DNA replication licensing factor MCM3	1.03	1.06	1.04	0.49	0.68	0.74	5.87 × 10^−8^
RIR1	Ribonucleoside-diphosphate reductase large subunit	0.98	1.19	1.16	0.43	0.84	0.71	5.66 × 10^−7^
RBM14	RNA-binding protein 14	1.15	0.97	1.08	0.38	0.44	0.60	8.30 × 10^−7^
RFA1	Replication protein A 70 kDa DNA-binding subunit	1.05	1.16	1.10	0.49	0.81	0.63	6.92 × 10^−6^
DUT	Deoxyuridine 5′-triphosphate nucleotidohydrolase	1.20	1.07	1.44	0.71	0.24	0.35	7.00 × 10^−6^
NP1L1	Nucleosome assembly protein 1-like 1	1.00	1.06	1.03	0.47	0.86	0.73	1.79 × 10^−5^
DPOE3	DNA polymerase epsilon subunit 3	1.10	0.89	1.10	0.56	0.37	0.46	5.71 × 10^−5^
PRI2	DNA primase large subunit	0.87	1.03	1.06	0.48	0.62	0.76	1.14 × 10^−4^
PURA	Transcriptional activator protein Pur-alpha	1.44	0.99	1.34	0.45	0.78	0.76	2.18 × 10^−4^
DPOLA	DNA polymerase alpha catalytic subunit	0.97	1.13	1.27	0.49	0.60	0.68	5.98 × 10^−4^
**Spliceosome**								
PCBP1	Poly(rC)-binding protein 1	0.92	0.88	0.94	0.38	0.51	0.60	2.55 × 10^−11^
RBM25	RNA-binding protein 25	0.97	0.95	1.07	0.49	0.43	0.45	6.81 × 10^−10^
DDX42	ATP-dependent RNA helicase DDX42	0.99	0.93	0.89	0.47	0.54	0.52	6.50 × 10^−9^
U5S1	116 kDa U5 small nuclear ribonucleoprotein component	1.01	0.98	0.82	0.38	0.54	0.58	4.54 × 10^−8^
RUXE	Small nuclear ribonucleoprotein E	0.91	1.92	2.19	0.26	1.17	1.39	1.55 × 10^−7^
THOC3	THO complex subunit 3	1.18	1.18	1.04	0.43	0.77	0.76	3.16 × 10^−7^
LSM8	U6 snRNA-associated Sm-like protein LSm8	1.09	1.22	1.42	0.24	0.79	0.88	4.85 × 10^−7^
DDX46	Probable ATP-dependent RNA helicase DDX46	0.96	1.16	1.08	0.45	0.68	0.66	5.52 × 10^−7^
SR140	U2 snRNP-associated SURP motif-containing protein	1.08	1.04	0.91	0.58	0.62	0.46	6.44 × 10^−7^
SPF27	Pre-mRNA-splicing factor SPF27	1.05	0.94	1.28	0.36	0.65	0.80	7.71 × 10^−7^
SNW1	SNW domain-containing protein 1	1.01	1.08	1.06	0.31	0.73	0.69	6.98 × 10^−5^
ROA3	Heterogeneous nuclear ribonucleoprotein A3	0.95	1.04	1.00	0.49	0.85	0.70	9.25 × 10^−5^
DDX23	Probable ATP-dependent RNA helicase DDX23	0.94	1.22	1.03	0.44	0.66	0.62	1.41 × 10^−4^
PR38B	Pre-mRNA-splicing factor 38B	0.54	1.17	1.37	0.32	0.94	1.09	2.42 × 10^−3^
**Carbon metabolism**								
PFKAM	ATP-dependent 6-phosphofructokinase muscle type	1.05	1.03	1.22	0.39	0.47	0.78	1.26 × 10^−8^
ENOA	Alpha-enolase	1.02	1.13	1.08	0.25	0.49	0.58	1.93 × 10^−8^
ACOC	Cytoplasmic aconitate hydratase	1.16	1.12	1.37	0.38	0.63	0.84	1.26 × 10^−7^
PFKAP	ATP-dependent 6-phosphofructokinase platelet type	1.01	1.06	1.18	0.34	0.46	0.70	2.83 × 10^−7^
IDH3A	Isocitrate dehydrogenase [NAD] subunit alpha	1.00	0.78	1.02	0.22	0.44	0.57	2.99 × 10^−7^
PGP	Glycerol-3-phosphate phosphatase	0.97	1.11	1.12	0.19	0.40	0.71	1.86 × 10^−6^
ODPB	Pyruvate dehydrogenase E1 component subunit beta mitochondrial	1.14	0.71	1.12	0.27	0.56	0.47	7.91 × 10^−6^
IDH3B	Isocitrate dehydrogenase [NAD] subunit beta	1.09	1.03	1.14	0.37	0.79	0.77	8.67 × 10^−6^
ALDOC	Fructose-bisphosphate aldolase C	1.03	1.28	1.25	0.49	1.07	1.09	2.17 × 10^−5^
MAOX	NADP-dependent malic enzyme	0.82	0.86	0.93	0.46	0.53	0.73	2.48 × 10^−4^

Color shading indicates to a value that has been downregulated or upregulated by 1.5 times or more. 1. 5 ≤ light green < 2.0, 2.0 ≤ green < 2.5, and 2.5 ≤ dark green for downregulation, and 1.5 ≤ light pink < 2.0 for upregulation.

**Table 3 biomolecules-11-00375-t003:** List of upregulated proteins from control and SiO_2_ NP groups of HepG2 cells by the exposure time associated with the biological pathway.

Protein	Description	Fold Change Compared to 0 h Control						*p*-Value
		Control (h)			SiO_2_ NPs (h)			
		4	10	16	4	10	16	
**Endosome**								
CD81	CD81 antigen	1.04	1.31	1.24	2.92	4.84	6.26	1.27 × 10^−9^
EGFR	Epidermal growth factor receptor	0.97	1.28	1.11	1.18	1.91	2.13	1.41 × 10^−9^
ANTR1	Anthrax toxin receptor 1	1.02	1.12	1.00	1.72	2.48	3.09	3.94 × 10^−8^
FOLR1	Folate receptor alpha	0.59	1.02	1.27	1.08	2.20	2.78	5.87 × 10^−8^
FLOT2	Flotillin-2	1.62	1.26	0.57	0.89	1.56	2.93	8.51 × 10^−7^
A4	Amyloid-beta precursor protein	0.87	0.87	0.99	1.37	2.07	1.83	1.45 × 10^−6^
TXND5	Thioredoxin domain-containing protein 5	0.94	1.70	1.49	1.82	2.23	2.01	1.62 × 10^−6^
LG3BP	Galectin-3-binding protein	1.07	1.31	1.44	1.27	2.03	2.49	2.01 × 10^−6^
B2MG	Beta-2-microglobulin	1.06	1.24	1.38	1.62	1.98	2.23	2.81 × 10^−6^
**Phagosome**								
HLAA	HLA class I histocompatibility antigen A alpha chain	0.94	1.20	1.22	1.25	2.23	2.01	1.36 × 10^−7^
ITB1	Integrin beta-1	1.06	1.15	1.22	0.75	1.90	2.39	3.53 × 10^−7^
ITAV	Integrin alpha-V	1.07	1.29	1.49	1.00	1.43	2.24	3.72 × 10^−7^
S61A1	Protein transport protein Sec61 subunit alpha isoform 1	1.00	1.33	1.48	0.64	1.19	2.38	2.94 × 10^−6^
VA0D1	V-type proton ATPase subunit d 1	1.02	1.63	1.28	0.94	1.94	2.16	1.81 × 10^−5^
**Lysosome**								
OST48	Dolichyl-diphosphooligosaccharide-protein glycosyltransferase 48 kDa subunit	0.88	1.42	1.23	2.11	2.84	3.07	2.08 × 10^−10^
CATA	Catalase	1.02	1.26	1.32	1.02	1.80	2.18	3.15 × 10^−10^
GNS	N-acetylglucosamine-6-sulfatase	0.97	1.39	1.75	0.71	1.82	2.56	1.04 × 10^−8^
BST2	Bone marrow stromal antigen 2	0.84	1.11	1.80	1.33	2.26	3.18	7.89 × 10^−8^
HM13	Minor histocompatibility antigen H13	0.95	1.24	1.19	1.40	2.11	2.42	1.06 × 10^−7^
SQSTM	Sequestosome-1	0.86	0.75	0.61	1.18	2.16	2.70	2.33 × 10^−7^
LAMP2	Lysosome-associated membrane glycoprotein 2	1.07	1.64	1.80	1.51	2.56	3.12	2.09 × 10^−6^
PCYOX	Prenylcysteine oxidase 1	1.12	1.26	1.40	2.35	2.50	2.40	3.10 × 10^−6^
NICA	Nicastrin	1.12	1.26	1.39	1.16	1.48	2.07	6.15 × 10^−5^
**Lipid metabolism**								
DEGS1	Sphingolipid delta(4)-desaturase DES1	0.99	0.99	0.88	1.79	1.88	2.09	4.79 × 10^−10^
ABCD3	ATP-binding cassette sub-family D member 3	1.03	1.40	1.31	1.17	1.86	3.00	1.69 × 10^−9^
MBOA7	Lysophospholipid acyltransferase 7	0.86	1.18	1.02	1.95	2.05	2.11	9.06 × 10^−9^
NCEH1	Neutral cholesterol ester hydrolase 1	1.03	1.22	1.12	1.90	3.09	2.76	1.18 × 10^−7^
CTL1	Choline transporter-like protein 1	1.22	1.80	1.25	1.07	3.86	3.63	1.93 × 10^−7^
ERLN2	Erlin-2	0.80	1.21	0.75	1.03	1.99	2.03	3.21 × 10^−7^
MGST3	Microsomal glutathione S-transferase 3	0.86	1.88	1.46	1.64	2.13	1.98	8.53 × 10^−7^
ACAD9	Complex I assembly factor ACAD9 mitochondrial	1.17	1.27	1.44	1.32	1.77	2.10	1.01 × 10^−6^
GPAA1	Glycosylphosphatidylinositol anchor attachment 1 protein	1.09	1.03	0.67	3.12	2.96	3.06	1.05 × 10^−6^
ACADM	Medium-chain specific acyl-CoA dehydrogenase mitochondrial	1.19	1.44	1.43	2.53	1.62	1.64	1.79 × 10^−6^
FDFT	Squalene synthase	0.89	1.23	1.75	1.00	1.56	2.37	2.84 × 10^−6^
ACOD	Acyl-CoA desaturase	0.86	1.57	1.01	2.45	2.10	1.31	6.59 × 10^−6^
HACL1	2-hydroxyacyl-CoA lyase 1	1.06	1.34	1.15	1.40	2.00	1.49	9.06 × 10^−3^

Color shading indicates to a value that has been downregulated or upregulated by 1.5 times or more. 1.5 ≤ light pink < 2.0, 2.0 ≤ pink < 2.5, and 2.5 ≤ red for upregulation, and 1.5 ≤ light green < 2.0 for downregulation.

## Data Availability

All nanoLC-ESI-MS/MS raw data and PEAKS search result were deposited to the ProteomeXchange Consortium via the PRIDE partner repository with the dataset identifier PXD024206.

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
