# Peer review of "Global Proteomics to Study Silica Nanoparticle-Induced Cytotoxicity and Its Mechanisms in HepG2 Cells"

_biomolecules, 2021, doi:10.3390/biom11030375_

Round 1

Reviewer 1 Report

The research presented in this manuscript is of great interest. The authors approached the molecular basis underlying cytotoxicity of SiO2 NP in the in vitro model (HepG2 cells), and through a MS-based quantitative proteomics approach.

Results achieved are clearly presented. Following a deep analysis of their data, authors were are able to propose a very comprehensive model of the cytotoxic events that could occur when hepatic cells are exposed to SiO2 NPs.

My recommendation is that the paper of Lee et al. is accepted.

Author Response

Thank you for your sincere and honest review. 

All reviewers' comments were grouped into one document, and I attached the file as a revision letter.

Thanks again for your time and review.

Best regards, 

JG Son

Reviewer 2 Report

The aim of the authors of the paper entitled “Global proteomics to study silica nanoparticle-induced 2 cytotoxicity and its mechanisms in HepG2 cells” was to provide insights into the cytotoxic mechanism induced by silica nanoparticles (SiO2 NP) in HepG2 cells.  To fulfil such a purpose they have performed proteomic analysis using mass spectrometry.

Based on the complex investigation it was found that the exposure time of cells to NP is important. Thus, if cell morphology was unchanged after 4 h, the protein expression involved in cell cycle, mitochondrial function or mRNA splicing was downregulated. After 10 h of treatment with SiO2 NP, a significant increase in ROS level was obtained. Also the level of proteins associated to intracellular oxidative phosphorylation and immune system was upregulated.

Finally, the order of biological events occurring at different time of exposure and the mechanisms underlying cytotoxic effect induced by SiO2 NP are suggestive schematic presented by the authors. 

The studies performed and the results obtained are very important giving more information about the safety and toxicity of NP widely used in nanomedicine. It is clear that protein profiling approach is a useful tool to analyse the cell response and to understand the cytotoxicity mechanism induced by NP.

In conclusion the manuscript is well written and organized and fits with the scope of the journal. 

Author Response

(The authors gave the same response as above.)

Reviewer 3 Report

The manuscript is clear and concise. The topic seems to be actual. However, the paper should be revised, major revision is recommended before publication.

Please briefly describe the process for the production of nanoparticles.

More detailed characterization should be carried out such as structure related measurements, BET, FTIR.

What about the stability?

It would be nice to see the evaluation of the percentage of apoptotic cells. The possibility that the nanoparticles elicit neuronal death via the involvement of an apoptotic process should be investigated.

Author Response

(The authors gave the same response as above.)

Round 2

Reviewer 3 Report

My recommendation is that the paper in this new form can be accepted